# Natural Sunlight-Mediated Emodin Photoinactivation of *Aeromonas hydrophila*

**DOI:** 10.3390/ijms25105444

**Published:** 2024-05-16

**Authors:** Gelana Urgesa, Liushen Lu, Jinwei Gao, Lichun Guo, Ting Qin, Bo Liu, Jun Xie, Bingwen Xi

**Affiliations:** 1Wuxi Fisheries College, Nanjing Agricultural University, Wuxi 214081, China; gelana.ur2018@gmail.com (G.U.); gaojinwei163@163.com (J.G.); 2Freshwater Fisheries Research Center, Chinese Academy of Fishery Sciences, Wuxi 214081, China; qint@ffrc.cn (T.Q.); liub@ffrc.cn (B.L.); xiej@ffrc.cn (J.X.); 3State Key Laboratory of Food Science and Technology, School of Food Science and Technology, Jiangnan University, Wuxi 214122, China; 8202306018@jiangnan.edu.cn

**Keywords:** *Aeromonas hydrophila*, emodin, photoinactivation, Chinese herbal medicine, reactive oxygen species

## Abstract

*Aeromonas hydrophila* can be a substantial concern, as it causes various diseases in aquaculture. An effective and green method for inhibiting *A. hydrophila* is urgently required. Emodin, a naturally occurring anthraquinone compound, was exploited as a photo-antimicrobial agent against *A. hydrophila*. At the minimum inhibitory concentration of emodin (256 mg/L) to inactivate *A. hydrophilia* in 30 min, an 11.32% survival rate was observed under 45 W white compact fluorescent light irradiation. In addition, the antibacterial activity under natural sunlight (0.78%) indicated its potential for practical application. Morphological observations demonstrated that the cell walls and membranes of *A. hydrophila* were susceptible to damage by emodin when exposed to light irradiation. More importantly, the photoinactivation of *A. hydrophila* was predominantly attributed to the hydroxyl radicals and superoxide radicals produced by emodin, according to the trapping experiment and electron spin resonance spectroscopy. Finally, a light-dependent reactive oxygen species punching mechanism of emodin to photoinactivate *A. hydrophila* was proposed. This study highlights the potential use of emodin in sunlight-mediated applications for bacterial control, thereby providing new possibilities for the use of Chinese herbal medicine in aquatic diseases prevention.

## 1. Introduction

*Aeromonas hydrophila* is an important bacterial pathogen that is associated with several diseases in aquatic animals, including hemorrhagic septicemia, fin and tail rot, epizootic ulcerative syndrome, and necrotizing skin and soft tissue infections, especially in immunocompromised individuals [1,2]. *A. hydrophila* can utilize virulence factors (including exotoxins, extracellular proteases, adhesins, etc.) to infect aquatic animals such as fish, shrimp, and crabs, leading to explosive epidemic diseases [3]. *A. hydrophila* exhibits broad pathogenicity, with a mortality rate of over 95% [4]. Moreover, it can cause a range of infections in humans, including skin and soft tissue infections, gastroenteritis, and sepsis [5]. Thus, inhibiting the growth of *A. hydrophila* facilitates the alleviation of various diseases caused by this bacterium. These *A. hydrophila*-induced diseases are usually treated with antibiotics. However, *A. hydrophila* is generally resistant to common pharmaceuticals and antibiotics such as penicillin, ampicillin, and colistin, which renders *A. hydrophila* inactivation difficult [6]. Moreover, abuse of antibiotics may pollute the environment and increase the risk of resistance in aquatic microorganisms [7]. Therefore, it is urgent and important to develop a novel, low-cost, environmentally friendly, and efficient method for inhibiting the growth of *A. hydrophila*.

Prominent among novel non-antibiotic approaches is likely to be the group of light-based technologies, including ultraviolet C irradiation therapy, photo-dynamic therapy, blue light therapy, and other light-based therapies [8,9,10,11]. In fact, photocatalysis is a promising approach to control microbial pathogen-induced infections owing to its green, efficient, and visible light utilization properties [12]. Logically, it is feasible to use photocatalysis to inactivate the aquatic pathogen *A. hydrophila*. The photoinactivation of pathogens using photosensitization can be explored as a potential technique for food safety [13]. The mechanism of this inactivation involves physical damage to the cell wall and potential cell lysis, which ultimately leads to microorganism death [14]. The compelling benefits of photoinactivation stem from its capacity to render microbes inactive, irrespective of antibiotic resistance [15]. Additionally, the possibility of microbes developing resistance to light-based inactivation is fundamentally low, as the targets involved are nonspecific [16].

The naturally occurring anthraquinone emodin can be easily acquired from many medicinal plants such as *Polygonum cuspidatum*, *Rheum offcinale*, and *Cassia obtusifolia*. It has been used as a traditional Chinese herb because of its antibacterial, antifungal, antiviral, and anti-inflammatory properties for more than 2000 years [17]. In addition, emodin exhibits an excellent photocatalytic activity for pollutant degradation and pathogen inhibition, owing to its large π-conjugation structure [18]. For example, emodin can damage the genomic DNA and membrane integrity of multidrug-resistant *Acinetobacter baumannii* under light irradiation [19]. Emodin has also been demonstrated to photoinactivate *Staphylococcus aureus* and *Escherichia coli* through the photogeneration of reactive oxygen species (ROS) [20]. It seems that photoinactivation may be an effective and alternative therapeutic option against the investigated bacteria [21]. However, the specific photoactivation of emodin’s bactericidal activity against *A. hydrophila* has not been directly studied. Therefore, we envisaged using the ability of emodin to photo-induce ROS to augment its antibacterial activity.

Herein, emodin as a photocatalyst can effectively photoinactivate the aquatic pathogen *A. hydrophila* within 30 min, upon 32 W white compact fluorescent light (CFL) irradiation. More importantly, *A. hydrophila* can be almost completely eradicated within 30 min under natural sunlight. Thus, this study dexterously combines the active ingredient in Chinese herbs to the inactivation of important pathogens in aquaculture and provides a new sight and possibility for achieving a practical application of Chinese herbal medicine in aquaculture.

## 2. Results

### 2.1. Visible Light Enhanced the Bactericidal Activity of Emodin

In this study, we examined the minimal inhibitory concentration (MIC) using the microbroth dilution method in a sterile U-bottom 96-well microplate. As expected, the negative control (DMSO) demonstrated no bacterial inhibition. The concentration of emodin ranged exponentially from 1 to 2048 mg/L, indicating the requirement of varying concentrations of emodin to effectively curb bacterial growth, as depicted in Figure 1. We found that the MIC of emodin against *A. hydrophila* NJ-35 was 256 mg/L. The results suggested that emodin inhibited bacterial growth in a concentration-dependent manner. Emodin concentrations below 256 mg/L allowed *A. hydrophila* growth, whereas those concentrations equal to or greater than 256 mg/L inhibited bacterial growth.

In order to determine whether light can enhance the inhibitory ability of emodin against *A. hydrophila*, the inhibitory effects of emodin against *A. hydrophila* under dark and light conditions were compared, as shown in Figure 2. Obviously, *A. hydrophila* can grow well in the presence or absence of emodin under the dark condition, as shown in Figure 2a,b, indicating that the effect of 256 mg/L emodin on *A. hydrophila* was almost negligible under dark conditions. However, a remarkable impact of emodin was observed under 32 W CFL irradiation, as shown in Figure 2c,d. The survival rates of *A. hydrophila* were rapidly decreased after irradiation for 15 min (from 97.89% to 67.89%), 30 min (from 92.44% to 13.76%), and 60 min (from 86.39% to 10.38%) compared to the dark condition, respectively. These results provided evidence for our hypothesis and demonstrated that light irradiation can increase the ability of emodin to inactivate *A. hydrophila*.

### 2.2. Photocatalytic Activity of Emodin for A. hydrophila Inactivation

Because the survival rates of *A. hydrophila* at 30 min and 60 min were similar, 30 min was selected as the light exposure time for further study (Figure 2d). In addition, the concentration of emodin and the light source are important factors during photocatalytic processes. Therefore, the effects of emodin concentration and light source for photoinactivation of *A. hydrophila* were analyzed, as shown in Figure 3. At an emodin concentration below the MIC, the survival rate of *A. hydrophila* decreased as the concentration of emodin increased (Figure 3a). In addition, the survival rate of *A. hydrophila* was closely related to the MIC (13.76%) and 1/2 MIC (16.72%). Thus, the optimum concentration of emodin was 128 mg/L (1/2 MIC). Moreover, the bactericidal activity of emodin against *A. hydrophila* was enhanced when the CFL lamp power increased (Figure 3b). This result also proves that light can elevate the antibacterial activity of emodin.

### 2.3. Practical Application under Sunlight

To evaluate the practical feasibility of the photoinactivation of *A. hydrophila* using emodin, the natural sunlight was used as the light source against *A. hydrophila*. A suspension of *A. hydrophila* (50 mL) with 32, 64, 128, or 256 mg/L emodin was exposed to sunlight (Figure 4a,b). *A. hydrophila* grow well under sunlight without the presence of emodin. However, *A. hydrophila* can be inactivated effectively using emodin at a concentration of both 128 mg/L and 256 mg/L under sunlight irradiation (the average solar irradiance was 109.32 mW/cm^2^, which was detected using an FZ-A irradiator from Beijing Normal University photoelectric instrument factory), and the survival rate of *A. hydrophila* using emodin at a concentration of 32 mg/L and 64 mg/L under sunlight was 4.81% and 1.57%, respectively. Moreover, a large-scale reaction was performed, as shown in Figure 4c,d. Considering the concentration of emodin and the efficiency of photoinactivation, 64 mg/L emodin was added to the 200 mL *A. hydrophila* suspension system under the sunlight. Encouragingly, the survival rate of *A. hydrophila* was 4.33%, closely related to the survival rate in the 50 mL system (1.57%), which revealed a predictable potential of emodin to induce microbial photoinactivation, as well as revealing its practical application. These results increase our comprehension of the possibility of using emodin in real-world situations, thereby emphasizing its suitability for continued investigation and advancement in practical applications.

### 2.4. Effects of Emodin on the Structure of A. hydrophila under Light Irradiation

To study the mechanism underlying the emodin-mediated photoinactivation of *A. hydrophila*, we observed the morphology of *A. hydrophila* after emodin treatment, under light irradiation. Surprisingly, emodin can result in morphological alterations after light irradiation, as shown in Figure 5. Scanning electron microscopy (SEM) images showed that *A. hydrophila* without treatment cells exhibited an intact cell structure with a well-stacked and smooth cell wall (Figure 5a,e). In addition, the morphology of *A. hydrophila* cells exposed to emodin under the dark conditions exhibited slight changes (Figure 5b). However, upon exposure to emodin in the presence of light, the cell morphology gradually transformed and the cell surface becomes rough and fuzzy (Figure 5f), demonstrating that the cell walls and membranes may be destroyed by emodin under light irradiation. On the other hand, *A. hydrophila* became shriveled after exposure to emodin in the dark condition, according to transmission electron microscopy (TEM) images (Figure 5d). Furthermore, a higher degree of leakage of cellular content was observed (Figure 5h), which resulted in irreversible damage to the cells. The death of cells was consequently induced during the photoinactivation process.

### 2.5. Reactive Species in Photoinactivation of A. hydrophila

To determine the mechanism of photoinactivation, the reactive species were detected using trapping experiment. Thus, isopropyl alcohol (IPA) and p-benzoquinone (PBQ) were used to scavenge hydroxyl radicals (·OH) and superoxide radicals (·O_2_^−^), respectively, during the photocatalysis process. As shown in Figure 6a, the addition of IPA lead to an increase in the survival rate of *A. hydrophila*, which indicates that ·OH plays an important role in the photoinactivation of *A. hydrophila*. However, the addition of PBQ resulted in *A. hydrophila* having difficulty surviving under dark conditions, possibly because of its cytotoxicity towards this bacterium. Consequently, the role of ·O_2_^−^ in photocatalysis remains uncertain. To further determine the active species in the photocatalytic process, we conducted electron spin resonance (ESR) spectroscopy to capture the signal of ·OH and·O_2_^−^ using 5,5-dimethyl-1-pyrroline-N-oxide (DMPO). As depicted in Figure 6b, no signal was observed under the dark condition, while the signal of DMPO–·OH with four typical characteristic peaks (red line) with a ratio of 1:2:2:1 was observed. This demonstrated that hydroxyl radicals play an important role in the photoinactivation of *A. hydrophila*, which is consistent with the results of the trapping experiment. In addition, the signal of DMPO–·O_2_^−^ with six typical characteristic peaks (blue line) was detected under the light condition, indicating the presence of superoxide radicals during the photocatalytic process. These results revealed that both ·O_2_^−^ and ·OH play a key role during the photocatalytic process.

## 3. Discussion

Many traditional Chinese herbs have been developed for inhibiting *A. hydrophila*, because they are widely available and environmentally friendly. For example, resveratrol [22,23], thymol [24], fraxetin [25], oridonin [26], magnolol [27], baicalin [28], and emodin [29] from traditional Chinese herbs display an inhibitory activity against *A. hydrophila*. In addition, extracts from *Ficus* leaves [30], *Moringa oleifera* leaves [31], the *Araucaria angustifolia* [32] seed coat, and *Hesperozygis ringens* [33] also have the ability to inhibit the growth of *A. hydrophila*. Furthermore, extracts such as essential oils (e.g., clove [34], rosemary [35], and cinnamon [36]) exhibit the activity of inhibiting *A. hydrophila*. However, the slow efficacy, difficulty in controlling the dosage, and lack of understanding of the action mechanisms of these traditional herbal medicines have limited their application in aquaculture. Nevertheless, photocatalysis is an alternative method to overcome these limitations due to its efficient, green, and visible light availability.

Although it has been proven that photocatalytic technology has tremendous potential to induce the inactivation of pathogens [37], there are only a few studies investigating the photoinactivation of aquatic pathogens, especially *A. hydrophila* [38]. In this study, we utilized the photosensitivity of emodin obtained from a traditional Chinese herb and applied it to the photoinactivation of *A. hydrophila*. Compared with other studies reported previously for *A. hydrophila* photoinactivation, as summarized in Table 1, the emodin used in this study exhibited an excellent photoinactivation efficiency with a lower power visible light source in a larger system. More importantly, *A. hydrophila* can be inactivated effectively under natural sunlight in a 200 mL system using emodin. This great efficiency may be attributed to the remarkable photoelectrochemical properties of emodin [29,39], which allows emodin to produce ROS rapidly under light irradiation.

The ROS generation mechanism using emodin depends on the light source and can be directly detected through trapping experiments and ESR spectroscopy analysis. The significant inhibitory effects of IPA under visible light indicated that ·OH radicals play a crucial role in driving the photobactericidal activity, because IPA is a scavenger of ·OH. In contrast, the inhibitory results of IPA under dark conditions were ineffective because of the presence of a light-dependent mechanism that generates ROS (Figure 6a). In addition, the ESR spectroscopy results further proved that ·OH is an important reactive species, along with ·O_2_^−^ (Figure 6b). Moreover, the morphology of *A. hydrophila* cells were observed in the presence and absence of visible light and the damage of the cell wall and membrane structure demonstrate that light-dependent ROS are crucial during the process of photoinactivation.

Based on the above results, the light-dependent ROS punching mechanism of emodin for the photoinactivation of *A. hydrophila* was proposed (Figure 7). Under the visible light condition, emodin absorbs an appropriate wavelength of light and reaches the first singlet excited state, ^1^Em*. And then ^1^Em* can react with H_2_O to produce ·OH and the reduced form of emodin (Em^•−^), in which Em^•−^ can react with O_2_ to yield ·O_2_^−^ [45]. These generated ROS are subsequently attached to the surface of *A. hydrophila*, causing damage to their cell walls and membranes. Furthermore, emodin can enter the cell interior through breaking down cell walls and membranes using ROS and continue producing ROS to destroy DNA and proteins to promote the photoinactivation of *A. hydrophila* [46].

## 4. Materials and Methods

### 4.1. Chemicals and Reagents

Emodin with a purity of 98% was purchased from the Solarbio Chemical Reagent Company (Beijing, China). Luria bertani (LB) and phosphate-buffered saline (PBS) buffer were also purchased from Solarbio life science (Beijing, China). NB medium was purchased from Qingdao Hope Biotechnology Co., Ltd (Qingdao, China). Agar was purchased from Biofroxx Company (Einhausen, Germany). Dimethyl sulfoxide (DMSO) was purchased from Shanghai Aladdin Biochemical Technology Co., Ltd (Shanghai, China). *A. hydrophila* NJ-35 was obtained from Prof. Yong-Jie Liu (College of Veterinary Medicine, Nanjing Agricultural University, Nanjing, China). Sodium chloride (NaCl), glutaraldehyde, IPA, and PBQ were purchased from Energy Chemical (Shanghai, China). DMPO was obtained from Shanghai Yuanye bio-technology Co., Ltd (Shanghai, China). A compact fluorescent lamp (CFL) was used as the light source for the experiments.

### 4.2. MIC of Emodin against A. hydrophila

The minimal inhibitory concentration (MIC) is an essential parameter that determines the lowest concentration of an antimicrobial agent that can prevent the visible growth of microorganisms after overnight incubation. The MIC analysis was conducted in a 96-well microtiter plate with LB medium, using the broth micro-dilution techniques and guideline procedures for aerobic testing [47]. *A. hydrophila* strains were first sub-cultured on nutrient agar and incubated at 28 °C for 24 h. After incubation on agar, single colonies from the plate were introduced into individual flasks containing sterile LB medium (50 mL) and were incubated in a shaking incubator at 28 °C for 24 h, to ensure that the bacterial concentration was approximately at 10^6^ colony-forming units/milliliter (CFU/mL). Then, 90 μL of the cell suspension were taken to the 96-well microtiter plate. The following step involved adding 10 μL of different concentrations of emodin. The final concentrations of emodin were 0, 1, 2, 4, 8, 16, 32, 64, 128, 256, 512, 1024, and 2048 mg/L. The plate was then incubated at 28 °C for 24 h. Negative and positive controls consisted of wells containing only DMSO and wells containing LB including bacteria, respectively. The growth of *A. hydrophila* was monitored by measuring the absorbance at 600 nm using an MK3 microcoder (Thermo Scientific, Waltham, MA, USA). The antibacterial activity test was performed in triplicate.

### 4.3. Photoinactivation of A. hydrophila Using Emodin

Overnight cultured *A. hydrophila* was inoculated in fresh NB medium and grown until the cell density reached 10^7^ CFU/mL and was used for the photoinactivation test. Following this, 50 mL of the as-obtained *A. hydrophila* suspension was added to a 250 mL flask, and the flasks were placed about 5 cm around the lamp, as shown in Figure 8. In addition, controls without light irradiation were also performed. These flasks were incubated at 28 °C for 30 min, then 100 µL of the solution was diluted 10^5^ times using 0.9% NaCl and was spread on NB plates. These plates were then incubated at 28 °C for 24 h. Following incubation, colony counts were conducted and the survival rates were calculated as follows: survival rate (%) = number of colonies in the experimental group/number of colonies in control group × 100%.

Photoinactivation experiments were carried out using different light sources (5 W, 15 W, 23 W, 32 W, and 45 W) as well as in darkness, with exposure times of 15, 30, and 60 min. Various concentrations of emodin, including 0, 1/16, 1/8, 1/4, 1/2, and 1 MIC, were tested against *A. hydrophila*. The antibacterial effects of emodin photoinactivation were evaluated based on the survival rate of *A. hydrophila*.

To explore the practical applications of emodin in aquaculture, we investigated the effectiveness of sunlight-mediated (East 120°15′48″, North 31°30′41″; temperatures on October 24, 26, and 30 and January 10, 11, and 15 were 27 °C, 28 °C, and 27 °C and 8 °C, 10 °C, and 12 °C at about 1 pm, respectively) photoinactivation of *A. hydrophila*. The experiment involved two distinct groups, of which one was subjected to natural sunlight with varying emodin concentrations (32, 64, 128, and 256 mg/L) for a duration of 30 min in a 50 mL system, and the other was the control group maintained under dark conditions. In addition, a 200 mL *A. hydrophila* suspension scaled-up system was investigated in a 500 mL flask.

### 4.4. Morphology Analysis

To analyze the structural changes in bacterial cells, SEM and TEM were observed using Hitachi SU8010 and Hitachi H-7650 (Hitachi, Tokyo, Japan), respectively [48,49]. Briefly, *A. hydrophila* cells were exposed to CFL light for 30 min in the presence of emodin and the untreated cells were chosen for the morphological assessment. The cells were collected using centrifugation at 12,000 rpm for 2 min, then washed using PBS buffer (0.1 M, pH 7.4). The collected cells were fixed in 2.5% glutaraldehyde and analyzed using SEM and TEM.

### 4.5. Study on Reactive Species of Photoinactivation

To confirm the effect of ROS on the photoinactivation of *A. hydrophila*, trapping experiments were performed using 128 mg of IPA and PBQ (reactive species scavengers), which were used to capture ·OH and·O_2_^−^, respectively [50]. The experiments were performed in an open flask at 28 °C with a 32 W CFL light source in the center. In the presence and absence of light, 128 mg IPA and PBQ were added to the flasks containing the bacterial strain and 256 mg/L emodin. The flask was then incubated in a shaker at 180 rpm for 30 min. At specified time intervals, 10 μL of the suspension was collected and diluted 10^5^ times using 0.9% NaCl. Subsequently, 100 μL of the diluted solution was transferred to NB plates, spread evenly, and incubated overnight. Following incubation, the colonies were counted and the survival rate was calculated.

The reactive species were further confirmed using an EMXplus 10/12 ESR instrument (Bruker, Karlsruhe, Germany). For this measurement, DMPO was used as the scavenging reagent. In the presence and absence of light, 128 mg DMPO was added to a flask containing the bacterial strain [51]. After the solution was subjected to irradiation with a 32 W CFL for 30 min, a 1 mL suspension was collected and used for ESR detection.

### 4.6. Statistical Analysis

All the obtained data were replicated independently three times and are expressed as mean ± standard error.

## 5. Conclusions

In conclusion, this study utilized the photosensitivity of the active ingredient emodin in traditional Chinese herbal medicines, which can efficiently achieve the photoinactivation of *A. hydrophila* (survival rate 13.76%) under 32 W CFL irradiation using 256 mg/L emodin. In addition, a lower concentration of emodin (64 mg/L) can inactivate *A. hydrophila* effectively (survival rate 4.33%) under natural sunlight, which indicated the great potential for practical application. Furthermore, photogenerated ROS, including ·OH and ·O_2_^−^, can damage the cell walls and membranes of *A. hydrophila*. Finally, the light-dependent ROS punching process was the photoinactivation mechanism of emodin to *A. hydrophila*. Photoinactivation of *A. hydrophila* using emodin is a promising method, which can provide a new direction for the application of traditional Chinese herbs in the inactivation of aquatic pathogens.

## Figures and Tables

**Figure 1 ijms-25-05444-f001:**
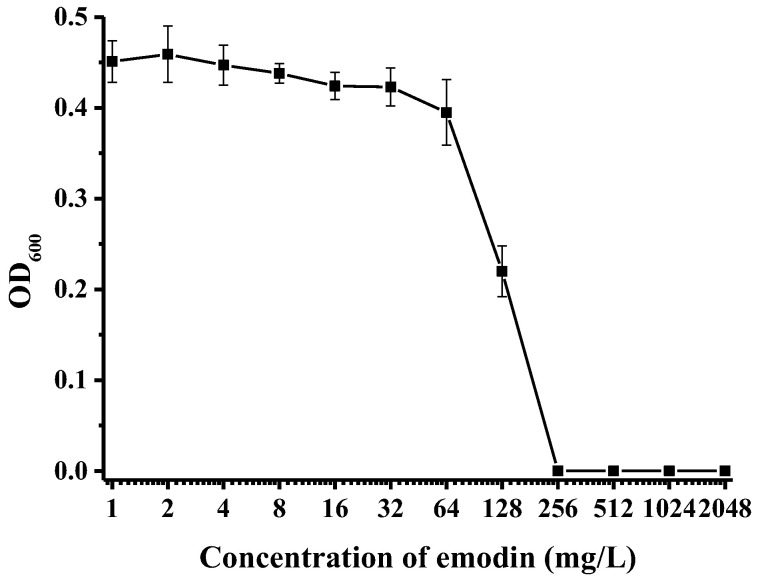
Minimal inhibitory concentration of emodin on the survival and growth curve of *A. hydrophila*.

**Figure 2 ijms-25-05444-f002:**
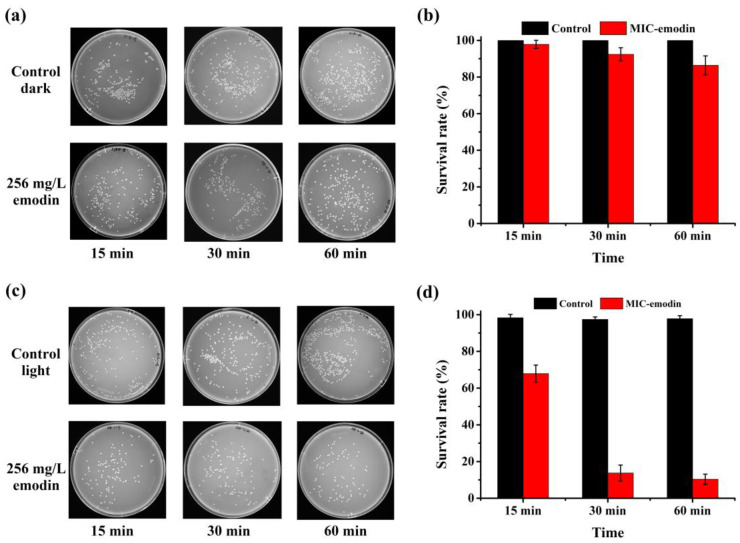
The activity of emodin under dark and light conditions. (**a**) The colony numbers of *A. hydrophila* on nutrient broth (NB) plates in the dark by 256 mg/L emodin. (**b**) The survival rate of *A. hydrophila* in the dark. (**c**) The colony numbers of *A. hydrophila* on NB plates upon 32 W CFL irradiation by 256 mg/L emodin. (**d**) The survival rate of *A. hydrophila* upon 32 W CFL irradiation.

**Figure 3 ijms-25-05444-f003:**
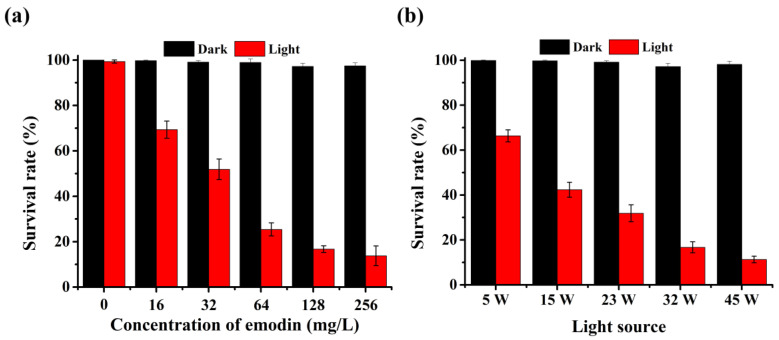
Effect of emodin concentration (**a**) and light source (**b**) for photoinactivation of *A. hydrophila*.

**Figure 4 ijms-25-05444-f004:**
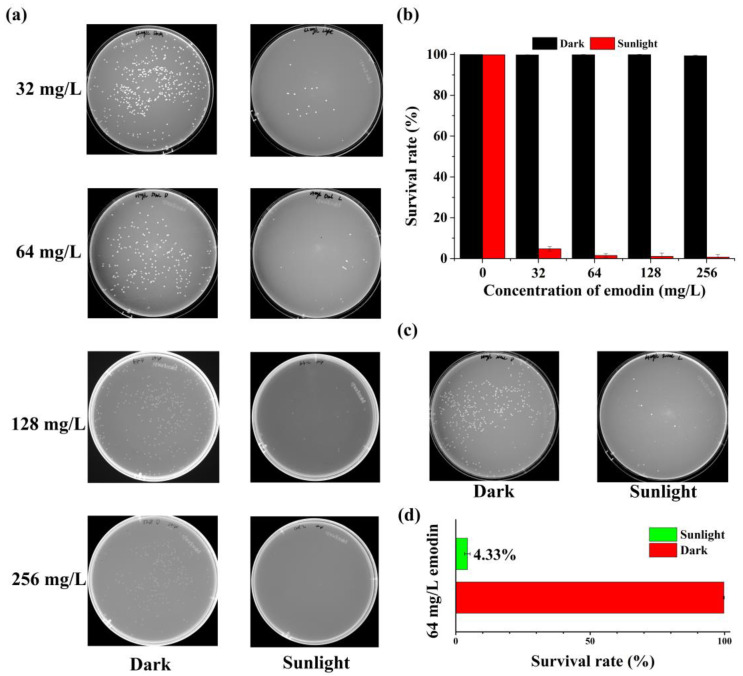
Photoinactivation of *A. hydrophila* under sunlight. (**a**) The colony numbers of *A. hydrophila* on NB plates under sunlight in a 50 mL system. (**b**) The survival rate of *A. hydrophila* under sunlight in a 50 mL system. (**c**) The colony numbers of *A. hydrophila* on NB plates under sunlight in a 200 mL system with 64 mg/L emodin. (**d**) The survival rate of *A. hydrophila* under sunlight in a 200 mL system with 64 mg/L emodin.

**Figure 5 ijms-25-05444-f005:**
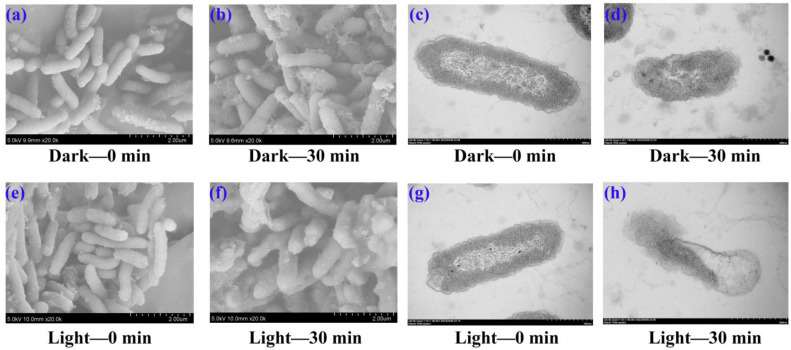
Effect of emodin on the structure of *A. hydrophila*. SEM images of *A. hydrophila* under dark conditions with 256 mg/L emodin treatment for 0 min (**a**) and 30 min (**b**). SEM images of *A. hydrophila* upon 32W CFL irradiation with 256 mg/L emodin treatment for 0 min (**e**) and 30 min (**f**). TEM images of *A. hydrophila* under dark conditions with 256 mg/L emodin treatment for 0 min (**c**) and 30 min (**d**). TEM images of *A. hydrophila* upon 32W CFL irradiation with 256 mg/L emodin treatment for 0 min (**g**) and 30 min (**h**).

**Figure 6 ijms-25-05444-f006:**
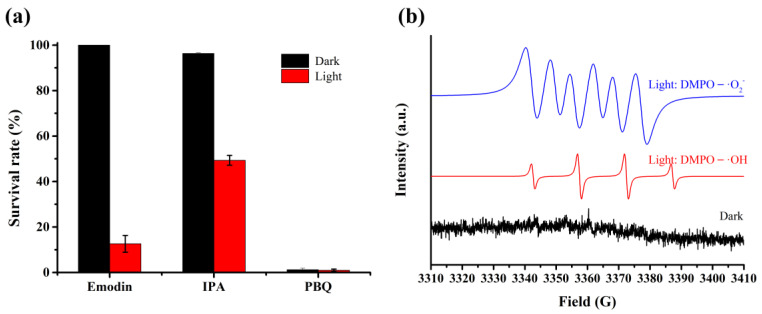
Trapping experiment of emodin for *A. hydrophila* photoinactivation in the presence of different scavengers (**a**) and ESR signal of DMPO–·O_2_^−^ and DMPO–·OH (**b**).

**Figure 7 ijms-25-05444-f007:**
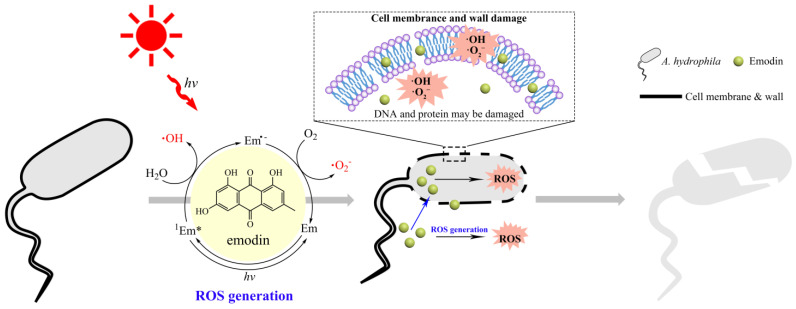
Proposed mechanism of sunlight-mediated bactericidal activity of emodin against *A. hydrophila*.

**Figure 8 ijms-25-05444-f008:**
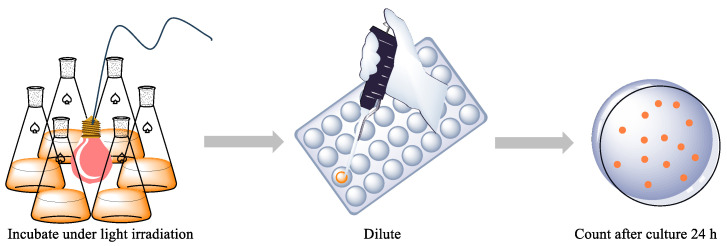
Schematic diagram of the antibacterial activity of emodin against *A. hydrophila*.

**Table 1 ijms-25-05444-t001:** Comparison of *A. hydrophila* photoinactivation based on different photocatalysts.

Photosensitizer	Concentration	Light Source	System	Irradiation Time	Efficiency Description	Reference
TiO_2_	20.5 g/m^2^	sunlight (980–1100 W/m^2^)	200 mL	2.5 min	1–1.4-fold decrease	[40]
erythrosine	0.01 mmol/L	green LED (130 mW/cm^2^)	500 μL	20 min	completely eradicated	[41]
erythrosine methyl ester	0.01 mmol/L	green LED (47 mW/cm^2^)	500 μL	30 min	survival rate about 24%	
erythrosine butyl ester	0.01 mmol/L	green LED (36 mW/cm^2^)	500 μL	30 min	survival rate about 7.7%	
curcumin	75 mmol/L	blue LED (232 mW/cm^2^)	500 μL	20 min	completely eradicated	[42]
curcumin	10 mg/L	18 W UV-A	5 mL	15 min	survival rate about 20%	[43]
pPdPc	8 μmol/L	LED (100 mW/cm^2^)	200 μL	15 min	completely eradicated	[44]
ZnPcMe	5 μmol/L	LED (100 mW/cm^2^)	200 μL	15 min	completely eradicated	
emodin	256 mg/L	32 W CFL (15.32 mW/cm^2^)	50 mL	30 min	survival rate 13.76%	this study
	64 mg/L	sunlight	50 mL	30 min	survival rate 1.57%	this study
	64 mg/L	sunlight	200 mL	30 min	survival rate 4.33%	this study

## Data Availability

Data is contained within the article and the corresponding author can make any materials available on request.

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
