# Peer review of "Natural Sunlight-Mediated Emodin Photoinactivation of Aeromonas hydrophila"

_ijms, 2024, doi:10.3390/ijms25105444_

Round 1
Reviewer 1 Report
Comments and Suggestions for Authors
The article "Natural sunlight-mediated emodin photoinactivation of Aeromonas hydrophila" describes the antibacterial activity of emodin under light irradiation. It is a valuable study that can be published after authors address the following problems:
Abstract should be checked and revised carefully by briefly introducing the work plan and key findings.
Abstracts should highlight the innovation of the article, as often abstract section is presented separately in search engines, it must be able to stand alone as an informative piece. In the abstract, need to focus more on the quantitative information, not qualitative one. The key quantitative data showing the antibacterial efficiency should be included in the abstract.
A new sub-section “statistical analysis” should be included at the end of section 4 providing the details of number of replicates, statistical analysis of significance and software used for statistical analysis with version and purchase details. Additionally, figures should include statistical significance details over the bars.
This work is interesting and can be boosted further. Hence the following literature could prove this manuscript doi: 10.3390/pharmaceutics15102470; doi: 10.3390/Ijms23116276; doi: 10.3390/pharmaceutics14122842; doi: 10.3390/Ijms24032657
Use uniform notation for measurement units (now for litre are used both l and L like at rows 253 or 255 but also elsewhere across the manuscript). Personally, I would recommend the use of L.
Figures 2a, 2c, 4 would be better if color instead of greyscale. Anyway a resolution improvement is a must for all figures.
Latin names of microbial strains must be italicized across the manuscript (figure 3, table 1, figure 7, figure 8 captions).
Row 155 missing verb (may be destroyed) or a rephrase is advisable.
Why authors used setups with 50 mL and 200 mL if the emodin used concentrations were the same? Authors should explain thoroughly the differences envisaged.
Row 190 “Chinese herbs such as resveratrol, thymol...”????
Abbreviations must be explained at first use (PBS row 292, explanation phosphate-buffered saline I suppose).
The conclusion should reflect the heuristic of the study. How is this system a better one? Conclusion section must be reworked to underline the novelty and advantages of this research, with actual numbers.
The conclusion part does not highlight the salient findings and future perspective
Author Response
Dear editor,
Thank you for considering our manuscript entitled “Natural sunlight-mediated emodin photoinactivation of Aeromonas hydrophila” for the publication in International Journal of Molecular Sciences. We greatly appreciate the comments and suggestions from the referees and the editors, which inspired us to further explore our investigation. As to some questions raised by the reviewers, we have the following replies and the replies were highlighted in blue. We also checked the grammar and typos carefully and revised our manuscript according to the reviewers’ opinions. The revised manuscript with changes using yellow highlights for easy identification was also prepared.
To Reviewer 1:
Q: The article "Natural sunlight-mediated emodin photoinactivation of Aeromonas hydrophila" describes the antibacterial activity of emodin under light irradiation. It is a valuable study that can be published after authors address the following problems:
A: We would like to thank the reviewer for careful and thorough reading of this manuscript and for the thoughtful comments and constructive suggestions, which help us to improve the quality of this manuscript. We have answered each of your points below.
Q1: Abstract should be checked and revised carefully by briefly introducing the work plan and key findings.
A1: Thanks for your advice. We have checked and revised carefully in the abstract. And the revised abstract has been attached as follows:
Aeromonas hydrophila can be a substantial concern as it causes various diseases in aquaculture. An effective and green method for inhibiting A. hydrophila is urgently required. Emodin, a naturally occurring anthraquinone compound, was exploited as a photo-antimicrobial agent against A. hydrophila. At the minimum inhibitory concentration of emodin (256 mg/L) to inactivate A. hydrophilia in 30 min, a 11.32% survival rate was observed under 45 W compact fluorescent light irradiation. In addition, the antibacterial activity under natural sunlight (0.78%), which indicated its potential for practical application. Morphological observations demonstrated that the cell walls and membranes of A. hydrophila were susceptible to damage by emodin when exposed to light irradiation. More importantly, the photoinactivation of A. hydrophila was predominantly attributed to the hydroxyl radicals and superoxide radicals produced by emodin, according to the trapping experiment and electron spin resonance spectroscopy. Finally, the light-dependent reactive oxygen species punching mechanism of emodin to photoinactivate A. hydrophila was proposed. This study highlights the potential use of emodin in sunlight-mediated applications for bacterial control, thereby providing new possibilities for the use of Chinese herbal medicine in aquatic diseases preventions.
Q2: Abstracts should highlight the innovation of the article, as often abstract section is presented separately in search engines, it must be able to stand alone as an informative piece. In the abstract, need to focus more on the quantitative information, not qualitative one. The key quantitative data showing the antibacterial efficiency should be included in the abstract.
A2: Thanks for your advice. We have added the detailed results data of the survival rate, emodin concentration, and the reaction concition in the abstract. And the revised abstract has been attached as follows:
Aeromonas hydrophila can be a substantial concern as it causes various diseases in aquaculture. An effective and green method for inhibiting A. hydrophila is urgently required. Emodin, a naturally occurring anthraquinone compound, was exploited as a photo-antimicrobial agent against A. hydrophila. At the minimum inhibitory concentration of emodin (256 mg/L) to inactivate A. hydrophilia in 30 min, a 11.32% survival rate was observed under 45 W compact fluorescent light irradiation. In addition, the antibacterial activity under natural sunlight (0.78%), which indicated its potential for practical application. Morphological observations demonstrated that the cell walls and membranes of A. hydrophila were susceptible to damage by emodin when exposed to light irradiation. More importantly, the photoinactivation of A. hydrophila was predominantly attributed to the hydroxyl radicals and superoxide radicals produced by emodin, according to the trapping experiment and electron spin resonance spectroscopy. Finally, the light-dependent reactive oxygen species punching mechanism of emodin to photoinactivate A. hydrophila was proposed. This study highlights the potential use of emodin in sunlight-mediated applications for bacterial control, thereby providing new possibilities for the use of Chinese herbal medicine in aquatic diseases preventions.
Q3: A new sub-section “statistical analysis” should be included at the end of section 4 providing the details of number of replicates, statistical analysis of significance and software used for statistical analysis with version and purchase details. Additionally, figures should include statistical significance details over the bars.
A3: Thanks for your advice. We have added statistical analysis section in the Materials and Methods. In addition, the differences between our groups were not very necessary, so we did not conduct a significance analysis of the data.
Q4: This work is interesting and can be boosted further. Hence the following literature could prove this manuscript doi: 10.3390/pharmaceutics15102470; doi: 10.3390/Ijms23116276; doi: 10.3390/pharmaceutics14122842; doi: 10.3390/Ijms24032657
A4: Thanks for your advice. We have added all sugggested references in the manuscript as ref. 8-11, 23.
Q5: Use uniform notation for measurement units (now for litre are used both l and L like at rows 253 or 255 but also elsewhere across the manuscript). Personally, I would recommend the use of L.
A5: Thanks for your advice. We have normalized all the units in the text.
Q6: Figures 2a, 2c, 4 would be better if color instead of greyscale. Anyway a resolution improvement is a must for all figures.
A6: Thanks for your advice. We had adjust the greyscale of figures 2a, 2c, and 4. In addition, the resolution of all figures were improved.
Q7: Latin names of microbial strains must be italicized across the manuscript (figure 3, table 1, figure 7, figure 8 captions).
A7: Thanks for your advice. We have italicized all the microbial strains in the text.
Q8: Row 155 missing verb (may be destroyed) or a rephrase is advisable.
A8: Thanks for your advice. We have revised it in the text.
Q9: Why authors used setups with 50 mL and 200 mL if the emodin used concentrations were the same? Authors should explain thoroughly the differences envisaged.
A9: Thanks for your advice. Because emodin at 256 mg/L can basically eradicate A. hydrophila under sunlight in 50 mL system, we optimized the emodin concentration under natural sunlight before scaling up the reaction. We found that a lower concentration of emodin (64 mg/L) can effectively inactivaion of A. hydrophila, thus the scaled-up reaction (200 mL) was conducted with 64 mg/L emodin.
Q10: Row 190 “Chinese herbs such as resveratrol, thymol...”????
A10: Thanks for your advice. We have revised this sentence as follows: resveratrol, thymol, fraxetin, oridonin, magnolol, baicalin and emodin from traditional Chinese herbs display the inhibitory activity against A. hydrophila.
Q11: Abbreviations must be explained at first use (PBS row 292, explanation phosphate-buffered saline I suppose).
A11: Thanks for your advice. We have explained all the abbreviations at first use.
Q12: The conclusion should reflect the heuristic of the study. How is this system a better one? Conclusion section must be reworked to underline the novelty and advantages of this research, with actual numbers.
A12: Thanks for your advice. We have added the detailed results data in conclusion. And the revised conclusion has been attached as follows.
In conclusion, this study utilized the photosensitivity of the active ingredient emodin in traditional Chinese herb, which can efficiently achieve the photoinactivation of A. hydrophila (survival rate 13.76%) under 32 W CFL irradiation by 256 mg/L emodin. In addition, a lower concentration of emodin (64 mg/L) can inactivate A. hydrophila effectively (survival rate 4.33%) under natural sunlight, which indicated the great potential for practical application. Furthermore, photogenerated ROS, including ·OH and ·O2-, can damage the cell walls and membranes of A. hydrophila. Finally, the light-dependent ROS punching process as the photoinactivation mechanism of emodin to A. hydrophila. Photoinactivation of A. hydrophila by emodin is a promising method, which can provide a new direction for the application of traditional Chinese herbs in the inactivation of aquatic pathogens.
Q13: The conclusion part does not highlight the salient findings and future perspective
A13: Thanks for your advice. We have highlighted the salient findings and future perspective in conclusion. And the revised conclusion has been attached as follows.
In conclusion, this study utilized the photosensitivity of the active ingredient emodin in traditional Chinese herb, which can efficiently achieve the photoinactivation of A. hydrophila (survival rate 13.76%) under 32 W CFL irradiation by 256 mg/L emodin. In addition, a lower concentration of emodin (64 mg/L) can inactivate A. hydrophila effectively (survival rate 4.33%) under natural sunlight, which indicated the great potential for practical application. Furthermore, photogenerated ROS, including ·OH and ·O2-, can damage the cell walls and membranes of A. hydrophila. Finally, the light-dependent ROS punching process as the photoinactivation mechanism of emodin to A. hydrophila. Photoinactivation of A. hydrophila by emodin is a promising method, which can provide a new direction for the application of traditional Chinese herbs in the inactivation of aquatic pathogens.
Reviewer 2 Report
Comments and Suggestions for Authors
The manuscript “Natural sunlight-mediated emodin photoinactivation of Aeromonas hydrophila” explores the idea of using natural compound emodin against the titled bacterium in aquaculture, by ROS generated upon light irradiation. This topic is interesting and worth exploring, however, the irradiation conditions are too questionable for the results obtained to be correctly evaluated and considered. For example, photoinactivation set-up as shown in Figure 8 does not seem to ensure measurable and continuously equal irradiation conditions. Full specifications of the used light source (compact fluorescent lamp) should be given in the Material and Methods, and not only power, but also irradiance, wavelengths and the applied light dose should be specified in all experiments with irradiation. Also, it would be useful to show UV/vis spectrum of emodin. Furthermore, in the experiments where sunlight was used, it seems that solar irradiance was not measured. Without average solar irradiance, the obtained results cannot be evaluated and compared (such as in Table 1).
Why among ROS only hydroxyl radical and superoxide radical anion were evaluated? What about singlet oxygen?
Subsection 4.5. does not mention emodin at all. Was it used, how and what was its concentration?
There is no statistical analysis in the Materials and Methods.
Line 169 – IPA and PBQ (also ESR in line 176 and DMPO in line 177) – first time mentioning – should be written with full names, then only abbreviations can be used.
Table 1 – entries 2-4 should be checked.
Comments on the Quality of English LanguageGrammar and style should be checked and corrected/revised throughout the text.
Figure 1, 2, 3 etc. caption - A. hydrophila – should be written in italics;
Figure 2 – ‘survival rate’ instead ‘survive rate’ (also in lines 116, 117, Table 1 etc.);
Line 65-66 – this sentence has to be revised;
Line 84/85 – demonstrated instead demonstrating;
Line 96 – ‘can grow’ (instead “can growth”);
Lines 180-182, 188-189, 204-206 – these sentences have to be revised;
Lines 316-317 - not clear, revise this sentence.
Author Response
Dear editor,
Thank you for considering our manuscript entitled “Natural sunlight-mediated emodin photoinactivation of Aeromonas hydrophila” for the publication in International Journal of Molecular Sciences. We greatly appreciate the comments and suggestions from the referees and the editors, which inspired us to further explore our investigation. As to some questions raised by the reviewers, we have the following replies and the replies were highlighted in blue. We also checked the grammar and typos carefully and revised our manuscript according to the reviewers’ opinions. The revised manuscript with changes using yellow highlights for easy identification was also prepared.
To Reviewer 2:
Q1: The manuscript “Natural sunlight-mediated emodin photoinactivation of Aeromonas hydrophila” explores the idea of using natural compound emodin against the titled bacterium in aquaculture, by ROS generated upon light irradiation. This topic is interesting and worth exploring, however, the irradiation conditions are too questionable for the results obtained to be correctly evaluated and considered. For example, photoinactivation set-up as shown in Figure 8 does not seem to ensure measurable and continuously equal irradiation conditions. Full specifications of the used light source (compact fluorescent lamp) should be given in the Material and Methods, and not only power, but also irradiance, wavelengths and the applied light dose should be specified in all experiments with irradiation. Also, it would be useful to show UV/vis spectrum of emodin. Furthermore, in the experiments where sunlight was used, it seems that solar irradiance was not measured. Without average solar irradiance, the obtained results cannot be evaluated and compared (such as in Table 1).
A1: We would like to thank the reviewer for careful and thorough reading of this manuscript and for the thoughtful comments and constructive suggestions, which help us to improve the quality of this manuscript. We have answered each of your points below.
(a) Alyhough photoinactivation set-up as shown in Figure 8 was difficult to ensure measurable and continuously equal irradiation conditions, this device ensures to a large extent that the lighting conditions of each flask are close based to the reference (J. Hazard. Mater. 2021, 419, 126555).
(b) We have added the information of the light source in the text. All the compact fluorescent lamp are white, and come from Philips. According to our previous study (Chem Eng J, 2021, 412, 128620), the irradiance of 32 W CFL lamp we used in the text is 15.32 mW/cm2, the light dose can be calculated as follows: light dose (J/m2) = irradiation time (s) × irradiance (W/m2). Therefore, the light dose of 32 W CFL lamp after 30 min irradiation is 27.58 J/cm2.
(c) UV/vis spectrum of emodin is an important factor, and we have detected it in previous work (Chemosphere, 2022, 305, 135401). However, the photosensitivity of emodin we used in this study has been reported, thus we did not include the UV/vis spectrum of emodin in the text.
(d) The solar irradiance is very important in this work, therefore, we detected the average solar irradiance as 109.32 mW/cm2.
Q2: Why among ROS only hydroxyl radical and superoxide radical anion were evaluated? What about singlet oxygen?
A2: Thanks for the reviewer to raise an important point here. Emodin can produce hydroxyl radicals and superoxide radicals by electron transfer, while singlet oxygen is formed by energy transfer. Indeed, in the trapping experiment, we use 1,4-diazabicyclooctane (DABCO) to capture singlet oxygen, but the additon of DABCO resulted in A. hydrophila difficult to survive in dark condition. In addition, we use 2,2,6,6-tetramethylpiperidine (TEMP) to capture singlet oxygen in ESR analysis, but we do not observed an obvious characteristic peak of singlet oxygen. Therefore, we cannot guarantee whether singlet oxygen is present in the photoinactivation process of A. hydrophila.
Q3: Subsection 4.5. does not mention emodin at all. Was it used, how and what was its concentration?
A3: Thanks for your advice. We use emodin in the study on reactive species of photoinactivation. The final concentration of emodin was 256 mg/L. We have added this detail in subsection 4.5.
Q4: There is no statistical analysis in the Materials and Methods.
A4: Thanks for your advice. We have added statistical analysis section in the Materials and Methods.
Q5: Line 169 – IPA and PBQ (also ESR in line 176 and DMPO in line 177) – first time mentioning – should be written with full names, then only abbreviations can be used.
A5: Thanks for your advice. We have revised them in the article. In addition, we checked all the abbreviation at the first time used.
Q6: Table 1 – entries 2-4 should be checked.
A6: Thanks for your advice. We have checked all the data described in the table based on these references.
Comments on the Quality of English Language
Q7: Grammar and style should be checked and corrected/revised throughout the text.
A7: Thanks for your advice. We have improved the language of the manuscript carefully to avoid grammatical, and bibliographic errors.
Q8: Figure 1, 2, 3 etc. caption - A. hydrophila – should be written in italics;
A8: Thanks for your advice. We have revised them in the article.
Q9: Figure 2 – ‘survival rate’ instead ‘survive rate’ (also in lines 116, 117, Table 1 etc.);
A9: Thanks for your advice. We have revised them in the article.
Q10: Line 65-66 – this sentence has to be revised;
A10: Thanks for your advice. We have revised this sentence as follows: Emodin has also been demonstrated to photoinactivate Staphylococcus aureus and Escherichia coli through the photogeneration of reactive oxygen species (ROS)
Q11: Line 84/85 – demonstrated instead demonstrating;
A11: Thanks for your advice. We have revised it in the article.
Q12: Line 96 – ‘can grow’ (instead “can growth”);
A12: Thanks for your advice. We have revised it in the article.
Q13: Lines 180-182, 188-189, 204-206 – these sentences have to be revised;
A13: Thanks for your advice. We have revised all these sentences in the article.
Q14: Lines 316-317 - not clear, revise this sentence.
A14: Thanks for your advice. We have revised this sentence as follows: In addition, a lower concentration of emodin (64 mg/L) can inactivate A. hydrophila effectively (survival rate 4.33%) under natural sunlight, which indicated the great potential for practical application.
Reviewer 3 Report
Comments and Suggestions for Authors
The manuscript "Natural sunlight-mediated emodin photoinactivation of Aeromonas hydrophila" is devoted to the possible use of chinese medical herb ingredient (emodin) for antimicrobial treatment of water to eradicate aquatic pathogen. While the topic is interesting, the work lacks novelty and clear practical significance.
Comments:
1. Antimicrobial and cytotoxic properties of emodin are well-known.
2. If emodin is proposed as antibiotic for the treatment of infection, it is unclear how photoactivation can be achieved in organism
3. As an alternative, if emodin is proposed as a reagent for water treatment, its applicability is also questionable. First of all, it is not obvious, how cytotoxic emodin can be removed after purification. Furthermore, there are photogenerators of ROS with significantly higher extinction coefficients and better photophysical properties. The advantages of emodin compared to well-established ROS-generating dyes are not obvious.
4. Emodin is known to produce also synglet oxygen under irradiation, but this effect was not quantified in this study.
Author Response
Dear editor,
Thank you for considering our manuscript entitled “Natural sunlight-mediated emodin photoinactivation of Aeromonas hydrophila” for the publication in International Journal of Molecular Sciences. We greatly appreciate the comments and suggestions from the referees and the editors, which inspired us to further explore our investigation. As to some questions raised by the reviewers, we have the following replies and the replies were highlighted in blue. We also checked the grammar and typos carefully and revised our manuscript according to the reviewers’ opinions. The revised manuscript with changes using yellow highlights for easy identification was also prepared.
To Reviewer 3:
Q: The manuscript "Natural sunlight-mediated emodin photoinactivation of Aeromonas hydrophila" is devoted to the possible use of chinese medical herb ingredient (emodin) for antimicrobial treatment of water to eradicate aquatic pathogen. While the topic is interesting, the work lacks novelty and clear practical significance.
A: We would like to thank the reviewer for careful and thorough reading of this manuscript and for the thoughtful comments and constructive suggestions, which help us to improve the quality of this manuscript. We have answered each of your points below.
Comments:
Q1: Antimicrobial and cytotoxic properties of emodin are well-known.
A1: Thanks for your advice. Although antimicrobial and cytotoxic properties of emodin are well-known, the antibacterial activity of emodin under dark condition was significantly lower than that under light condition (Fig. 2). In addition, the application of emodin photosensitivity has been reported rarely. Hence, utilizing the photosensitive properties of emodin to achieve photoinactivation of the pathogenic A. hydrophila in aquatic presents significant exploration opportunities and research value. This study provides important references for the development of antibacterial agents, with the potential to break through the dilemma of bacterial resistance. Additionally, it offers theoretical basis and application foundation for green aquatic drugs and precise disease prevention and control technologies.
Q2: If emodin is proposed as antibiotic for the treatment of infection, it is unclear how photoactivation can be achieved in organism
A2: Thanks for the reviewer to raise an important point here. Achieving light activation in organisms is not only a challenge for emodin in infection treatment, but also a common issue for photo-antimicrobial agents. Light permeability is crucial for photodynamic therapy, limiting its effectiveness to areas accessible to light.
Q3: As an alternative, if emodin is proposed as a reagent for water treatment, its applicability is also questionable. First of all, it is not obvious, how cytotoxic emodin can be removed after purification. Furthermore, there are photogenerators of ROS with significantly higher extinction coefficients and better photophysical properties. The advantages of emodin compared to well-established ROS-generating dyes are not obvious.
A3: Thanks for your advice. The MIC of emodin to A. hydrophila is high (256 mg/L) based on the Fig. 1. Therefore, the use of lower concentrations of emodin (64 and 128 mg/L) under light conditions has negligible cytotoxicity in this study. Although there are photogenerators of ROS with significantly higher extinction coefficients and better photophysical properties, they may have the disadvantages of high cost, harsh reaction conditions and high toxicity compared with emodin (Chem Rev, 2012, 112, 5520-5551; Ultrason Sonochem, 2019, 58, 104702; J Am Chem Soc, 2021, 143, 17891-17909). In addition, compared to well-established ROS-generating dyes, emodin is more available and has lower toxicity and higher biocompatibility as it is a natural product from Chinese herbal sources (Chemosphere, 2022, 305, 135401). Moreover, the photoinactivation efficiency of these well-established ROS-generating dyes against A. hydrophila remains unknown, as there has been little research in this area (Table 1). More importantly, A. hydrophila can be inactivated effectively by emodin at a lower concentration under natural sunlight irradiation in a larger system (Fig.4). Therefore, emodin has a good potential for practical application in water treatment.
Q4: Emodin is known to produce also synglet oxygen under irradiation, but this effect was not quantified in this study.
A4: Thanks for your advice. Although emodin can produce ROS under irradiation, these ROS are short-lived and difficult to capture. In addition, we can semi-quantitatively analyze the amount of ROS through ESR as shown in Fig. 6b.
Reviewer 4 Report
Comments and Suggestions for Authors
Abstract
11: Remove “can be a significant concern as it”
12: Write A. hydrophila in full
13: Remove “Learning from nature”
16, 22: Delete the abbreviation CSL and ESR as they were used only once, Write ROS in full
The structuring of the abstract needs to be revised: Aim of the study, methodology, result and conclusion
Introduction
46: Delete space in “photo- dynamic”
53-54: Provide reference for this claim “The compelling benefits of photoinactivation stem from its capacity to render microbes inactive, irrespective of antibiotic resistance”
59: Provide the English name for these plants in bracket “Polygonum cuspidatum, Rheum offcinale and Cassia obtusifolia”
58-59, 65-66, 151-152, 172-173, 181, 188-189, 253-255, 248-260, 262-270: These sentences are incomprehensible; recast them
Result
81-84, 126-128: Should be in the methodology not result
84-85,156, 262-270: The sentences should be reported in the past tense
87: It was observed that the MIC of emodin that inhibited A. hydrophila NJ-35 was 256 mg/L. This concentration seems high; therefore, the authors should provide the basis of categorizing emodin as effective antimicrobial agent. Previous works on antibacterial activity of emodin would contain the basis (MIC) for classifying emodin as a good candidate for the development of a novel antibacterial agent.
86-91: Should be in discussion section
93: MIC should be written in full
94-96, 102-104, 135-139, 174, 178-180, 182-183: Should be in the discussion section
97: Check the grammar “as show in”
106-109, 141-145, 171, Table 1: is it survive or survival rate? There were undefined abbreviations such as NB in the title and the name of organism was not italicized
111-114: should be in the methodology
129-130: Check the use of tenses “inactivate” “survive”
150, 169, 176: Give the full meaning of SEM, IPA, PBQ, ESR, DMP.OH the first time used
155: something is missing in the sentence
Discussion
190-191: The antibacterial compounds listed are not Chinese herbs, they are compounds from herds
188-195, 200-203: Should be in the introduction section/justification of study
I suggest table contains only the result of the substances used in the current study, and then the results compared with the previous studies in the table in the discussion section.
233: Mechanism action or Mechanism of action? A. hydrophila should be italicized
Methodology
The A. hydrophila strain used in the study should be described (source, genomic charcateristics) for reprodcucibility
236-247: The 4.1 section titled “Chemicals and reagents” should be collapsed into the methodology. For example, microbroth dilution was done using Luria Broth (Solarbio Life Science, China). Use the name of the manufacturing company not the vendor
227: Check double reference
248-260: There were no positive and negative control for the MIC test, and no quality-control strain of bacteria was used
306: Check spelling of Germany
309: What do you mean by “…was collected and detected in ESR”
Conclusion
311: Delete “ingeniously”
The authors should state the limitations of this study
Comments on the Quality of English LanguageModerate English editing required
Author Response
Dear editor,
Thank you for considering our manuscript entitled “Natural sunlight-mediated emodin photoinactivation of Aeromonas hydrophila” for the publication in International Journal of Molecular Sciences. We greatly appreciate the comments and suggestions from the referees and the editors, which inspired us to further explore our investigation. As to some questions raised by the reviewers, we have the following replies and the replies were highlighted in blue. We also checked the grammar and typos carefully and revised our manuscript according to the reviewers’ opinions. The revised manuscript with changes using yellow highlights for easy identification was also prepared.
To Reviewer 4:
Abstract
Q1: 11: Remove “can be a significant concern as it”
A1: Thanks for your advice. We have revised this sentence as follows: Aeromonas hydrophila can be a substantial concern as it causes various diseases in aquaculture.
Q2: 12: Write A. hydrophila in full
A2: Thanks for your advice. We had write Aeromonas hydrophila in full when it appears first in Line 11.
Q3: 13: Remove “Learning from nature”
A3: Thanks for your advice. We have removed “Learning from nature”.
Q4: 16, 22: Delete the abbreviation CSL and ESR as they were used only once, Write ROS in full
A4: Thanks for your advice. We have revised them in the article.
Q5: The structuring of the abstract needs to be revised: Aim of the study, methodology, result and conclusion
A5: Thanks for your advice. We have added the detailed results data of the survival rate, emodin concentration, and the reaction concition in the abstract. And the revised abstract has been attached as follows:
Aeromonas hydrophila can be a substantial concern as it causes various diseases in aquaculture. An effective and green method for inhibiting A. hydrophila is urgently required. Emodin, a naturally occurring anthraquinone compound, was exploited as a photo-antimicrobial agent against A. hydrophila. At the minimum inhibitory concentration of emodin (256 mg/L) to inactivate A. hydrophilia in 30 min, a 11.32% survival rate was observed under 45 W compact fluorescent light irradiation. In addition, the antibacterial activity under natural sunlight (0.78%), which indicated its potential for practical application. Morphological observations demonstrated that the cell walls and membranes of A. hydrophila were susceptible to damage by emodin when exposed to light irradiation. More importantly, the photoinactivation of A. hydrophila was predominantly attributed to the hydroxyl radicals and superoxide radicals produced by emodin, according to the trapping experiment and electron spin resonance spectroscopy. Finally, the light-dependent reactive oxygen species punching mechanism of emodin to photoinactivate A. hydrophila was proposed. This study highlights the potential use of emodin in sunlight-mediated applications for bacterial control, thereby providing new possibilities for the use of Chinese herbal medicine in aquatic diseases preventions.
Introduction
Q6: 46: Delete space in “photo- dynamic”
A6: Thanks for your advice. We have deleted the space.
Q7: 53-54: Provide reference for this claim “The compelling benefits of photoinactivation stem from its capacity to render microbes inactive, irrespective of antibiotic resistance”
A7: Thanks for your advice. We have added the reference in the manuscript as ref. 15.
Q8: 59: Provide the English name for these plants in bracket “Polygonum cuspidatum, Rheum offcinale and Cassia obtusifolia”
A8: Thanks for your advice. We have provided the English name for these plants in bracket.
Q9: 58-59, 65-66, 151-152, 172-173, 181, 188-189, 253-255, 248-260, 262-270: These sentences are incomprehensible; recast them
A9: Thanks for your advice. We have rewrited these sentences in the article.
Result
Q10: 81-84, 126-128: Should be in the methodology not result
A10: Thanks for your advice. We have moved these sentences to the methodology section.
Q11: 84-85,156, 262-270: The sentences should be reported in the past tense
A11: Thanks for your advice. We have revised them in the article.
Q12: 87: It was observed that the MIC of emodin that inhibited A. hydrophila NJ-35 was 256 mg/L. This concentration seems high; therefore, the authors should provide the basis of categorizing emodin as effective antimicrobial agent. Previous works on antibacterial activity of emodin would contain the basis (MIC) for classifying emodin as a good candidate for the development of a novel antibacterial agent.
A12: Thanks for your advice. In this study, we utilized the photosensitivity of emodin and applied this properity to photoinactivation of A. hydrophila. Although the MIC of emodin that inhibited A. hydrophila NJ-35 seems high, the photoinactivation efficiency of A. hydrophila is amazing according to Fig.2. In addition, the MIC of emodin that inhibited A. hydrophila NJ-35 under light condition is 8 mg/L, much lower than that under dark condition (256 mg/L). This results demonstrated that emodin can be a good candidate for the development of a novel photo-antimicrobial agent.
Q13: 86-91: Should be in discussion section
A13: Thanks for your advice. According to the Fig.1, we can observe that emodin concentrations below 256 mg/L allowed A. hydrophila growth, whereas those concentrations equal to or greater than 256 mg/L inhibited bacterial growth. Therefore, this sentence was in the results section.
Q14: 93: MIC should be written in full
A14: Thanks for your advice. We had write it in full.
Q15: 94-96, 102-104, 135-139, 174, 178-180, 182-183: Should be in the discussion section
A15: Thanks for your advice. These sentences are used to describe the results of the experiment, so these sentences are more suitable for the results section.
Q16: 97: Check the grammar “as show in”
A16: Thanks for your advice. We have changed it as follows: as shown in.
Q17: 106-109, 141-145, 171, Table 1: is it survive or survival rate? There were undefined abbreviations such as NB in the title and the name of organism was not italicized
A17: Thanks for your advice. We have changed “survive rate” to “survival rate” in the text. We have checked all the abbreviations at first use and explained them.
Q18: 111-114: should be in the methodology
A18: Thanks for your advice. These sentences described the results of Fig.2 and 3, so these sentences are more suitable for the results section.
Q19: 129-130: Check the use of tenses “inactivate” “survive”
A19: Thanks for your advice. We have checked and revised them.
Q20: 150, 169, 176: Give the full meaning of SEM, IPA, PBQ, ESR, DMP.OH the first time used
A20: Thanks for your advice. We had give the full meaning of SEM, IPA, PBQ, and DMPO at the first time used. In addition, we checked all the abbreviation at the first time used.
Q21: 155: something is missing in the sentence
A21: Thanks for your advice. We have rewrited these sentences as follows: However, upon exposure to emodin in the presence of light, the cell morphology gradually transformed, and cell surface becomes rough and fuzzy (Fig. 5f), demonstrating that the cell walls and membranes may be destroyed by emodin under light irradiation.
Discussion
Q22: 190-191: The antibacterial compounds listed are not Chinese herbs, they are compounds from herds
A22: Thanks for your advice. We have rewrited these sentences as follows: For example, resveratrol, thymol, fraxetin, oridonin, magnolol, baicalin and emodin from traditional Chinese herbs display the inhibitory activity against A. hydrophila.
Q23: 188-195, 200-203: Should be in the introduction section/justification of study
A23: Thanks for your advice. Although these sentences can be placed in the introduction section/justification of study, these sentences have more important significance for the discussion of the mechanism, so they are more suitable for the discussion section.
Q24: I suggest table contains only the result of the substances used in the current study, and then the results compared with the previous studies in the table in the discussion section.
A24: Thanks for your advice. In order to compare the photoinactivation efficiency of A. hydrophila with that of previous studies more clearly, the efficiency of emodin photoinactivation of A. hydrophila in this study was placed in the same table with previous studies.
Q25: 233: Mechanism action or Mechanism of action? A. hydrophila should be italicized
A25: Thanks for your advice. We have revised it as follows: Mechanism of sunlight-mediated bactericidal activity of emodin against A.hydrophila.
Methodology
Q26: The A. hydrophila strain used in the study should be described (source, genomic charcateristics) for reprodcucibility
A26: Thanks for your advice. In this study, the strain we used was A. hydrophila NJ-35, which was obtained from Prof. Yong-Jie Liu (College of Veterinary Medicine, Nanjing Agricultural University, Nanjing, China).
Q27: 236-247: The 4.1 section titled “Chemicals and reagents” should be collapsed into the methodology. For example, microbroth dilution was done using Luria Broth (Solarbio Life Science, China). Use the name of the manufacturing company not the vendor
A27: Thanks for your advice. The 4.1 section was collapsed into the methodlogy, and we instead vendor to the manufacturing company.
Q28: 227: Check double reference
A28: Thanks for your advice. We have checked all the references in the text carefully.
Q29: 248-260: There were no positive and negative control for the MIC test, and no quality-control strain of bacteria was used
A29: Thanks for your advice. We have added this details in the text as follows: Negative and positive controls consisted of wells containing only DMSO and wells containing LB including bacteria, respectively.
Q30: 306: Check spelling of Germany
A30: Thanks for your advice. We have revised it.
Q31: 309: What do you mean by “…was collected and detected in ESR”
A31: Thanks for your advice. We described this sentence as follows: After the solution was subjected to irradiation with a 32 W CFL for 30 min, and 1 mL suspension was collected and used for ESR detection.
Conclusion
Q32: 311: Delete “ingeniously”
A32: Thanks for your advice. We deleted “ingeniously” in the text.
Q33: The authors should state the limitations of this study
A33: Thanks for your advice. Achieving light activation in organisms is not only a challenge for emodin in infection treatment, but also a common issue for photo-antimicrobial agents. Light permeability is crucial for photodynamic therapy, limiting its effectiveness to areas accessible to light.
Q34: Moderate English editing required
A34: Thanks for your advice. We have improved the language of the manuscript carefully to avoid grammatical, and bibliographic errors.
Round 2
Reviewer 2 Report
Comments and Suggestions for Authors
The authors improved the manuscript and answered certain questions and concerns. However, as they themselves confirmed, sunlight is very important for this research, thus the conditions for these experiments must be very clear. Sunlight intensity data are still not entered in the manuscript, and also Materials and methods section lack data on the instrumentation used to measure average solar irradiance. Furthermore, it is very important to show control where the bacteria were exposed to sunlight without the presence of emodin to check the effect of 'light only' on the bacterium. In the section 2.3. there is no 'sunlight only' control.
Comments on the Quality of English LanguageThere are still some grammatical errors and typos (e.g. line 60).
Author Response
Dear editor,
Thank you for considering our manuscript entitled “Natural sunlight-mediated emodin photoinactivation of Aeromonas hydrophila” for the publication in International Journal of Molecular Sciences. We greatly appreciate the comments and suggestions from the referees and the editors, which inspired us to further explore our investigation. As to some questions raised by the reviewers, we have the following replies and the replies were highlighted in blue. We also checked the grammar and typos carefully and revised our manuscript according to the reviewers’ opinions. The revised manuscript with changes using green highlights for easy identification was also prepared.
To Reviewer 2:
Q1: The authors improved the manuscript and answered certain questions and concerns. However, as they themselves confirmed, sunlight is very important for this research, thus the conditions for these experiments must be very clear. Sunlight intensity data are still not entered in the manuscript, and also Materials and methods section lack data on the instrumentation used to measure average solar irradiance. Furthermore, it is very important to show control where the bacteria were exposed to sunlight without the presence of emodin to check the effect of 'light only' on the bacterium. In the section 2.3. there is no 'sunlight only' control.
A1: Thanks for your advice. We have added the sunlight intensity data in the section 2.3, including the average solar irradiance and the instrumentation. In addition, A. hydrophila can grow well under 32 W CFL irradiation without emodin as shown in Fig. 2c. Therefore, we believe that A. hydrophila can grow well under sunlight without emodin. In fact, we investigated this experiment on May 6 as shown in following figure, obviously, A. hydrophila can grow well at the presence or absence of sunlight without emodin. Thus, we added this results in Fig 4b.
Q2: There are still some grammatical errors and typos (e.g. line 60).
A2: Thanks for your advice. We have revised this sentence as follows: It has been used as a traditional Chinese herb because of its antibacterial, antifungal, antiviral and anti-inflammatory properties for more than 2000 years. In addition, we have carefully checked to avoid grammatical and bibliographic errors.

Reviewer 3 Report
Comments and Suggestions for Authors
The revised version of the manuscript contains no additional data regarding major comments in the original review report. The authors reply on the significance of the finding remains rather speculative. Emodin is clearly not suitable for the treatment of the infection due to photosensitizing mode of action. Hypothetical advantages of emodin compared with other photoinactivators in water purification should be supported by experimental comparison, including the control of practically significant parameters of water after treatment.
Author Response
Dear editor,
Thank you for considering our manuscript entitled “Natural sunlight-mediated emodin photoinactivation of Aeromonas hydrophila” for the publication in International Journal of Molecular Sciences. We greatly appreciate the comments and suggestions from the referees and the editors, which inspired us to further explore our investigation. As to some questions raised by the reviewers, we have the following replies and the replies were highlighted in blue. We also checked the grammar and typos carefully and revised our manuscript according to the reviewers’ opinions. The revised manuscript with changes using green highlights for easy identification was also prepared.
To Reviewer 3:
Q: The revised version of the manuscript contains no additional data regarding major comments in the original review report. The authors reply on the significance of the finding remains rather speculative. Emodin is clearly not suitable for the treatment of the infection due to photosensitizing mode of action. Hypothetical advantages of emodin compared with other photoinactivators in water purification should be supported by experimental comparison, including the control of practically significant parameters of water after treatment.
A: Thanks for your advice. Although we do not have experimental data indicating that emodin has advantages in water purification compared to other photoinactivators, emodin, as a photo-antimicrobial agent derived from traditional Chinese medicine, has certain advantages in inactivating A. hydrophila as shown in Table 1, the emodin used in this study exhibited excellent photoinactivation efficiency with a lower power visible light source in a larger system. More importantly, A. hydrophila can be inactivated effectively under natural sunlight in a 200 mL system by emodin. In addition, for the applicability for treating infections, photo-antimicrobial agents can only play a role in the place where light can reach, emodin as a natural photo-antimicrobial agent, it is difficult to play a role in organisms. However, it has application potential in shallow water, equipment surface and aquaculture water. Moreover, the application of photo-antimicrobial agents in aquaculture still requires a lot of research and development work to ensure their effectiveness, safety, and scalability.
Table 1. Comparation of A. hydrophila photoinactivation based on different photocatalysts.
|
Photosensitizer |
Concentration |
Light source |
System |
Irradiation time |
Efficiency description |
Reference |
|
TiO2 |
20.5 g/m2 |
sunlight (980-1100 W/m2) |
200 mL |
2.5 min |
1 - 1.4-fold decrease |
[40] |
|
erythrosine erythrosine methyl ester erythrosine butyl ester |
0.01 mmol/L 0.01 mmol/L 0.01 mmol/L |
green LED (130 mW/cm2) green LED (47 mW/cm2) green LED (36 mW/cm2) |
500 μL 500 μL 500 μL |
20 min 30 min 30 min |
completely eradicated survival rate about 24% survival rate about 7.7% |
[41] |
|
curcumin |
75 mmol/L |
blue LED (232 mW/cm2) |
500 μL |
20 min |
completely eradicated |
[42] |
|
curcumin |
10 mg/L |
18 W UV-A |
5 mL |
15 min |
survival rate about 20% |
[43] |
|
pPdPc ZnPcMe |
8 μmol/L 5 μmol/L |
LED (100 mW/cm2) LED (100 mW/cm2) |
200 μL 200 μL |
15 min 15 min |
completely eradicated completely eradicated |
[44] |
|
emodin |
256 mg/L 64 mg/L 64 mg/L |
32 W CFL (15.32 mW/cm2) sunlight (109.32 mW/cm2) sunlight (109.32 mW/cm2) |
50 mL 50 mL 200 mL |
30 min 30 min 30 min |
survival rate 13.76% survival rate 1.57% survival rate 4.33% |
this study this study this study |

Round 3
Reviewer 2 Report
Comments and Suggestions for Authors
The authors have addressed the questions raised in the review and supplemented the manuscript, therefore I suggest publication of the paper.
Author Response
Dear editor,
Thank you for considering our manuscript entitled “Natural sunlight-mediated emodin photoinactivation of Aeromonas hydrophila” for the publication in International Journal of Molecular Sciences. We greatly appreciate the comments and suggestions from the referees and the editors, which inspired us to further explore our investigation. As to some questions raised by the reviewers, we have the following replies and the replies were highlighted in blue. We also checked the grammar and typos carefully and revised our manuscript according to the reviewers’ opinions. The revised manuscript with changes using red highlights for easy identification was also prepared.
To Reviewer 2:
Q: The authors have addressed the questions raised in the review and supplemented the manuscript, therefore I suggest publication of the paper.
A: We would like to thank the reviewer for careful and thorough reading of this manuscript and for the thoughtful comments and constructive suggestions, which help us to improve the quality of this manuscript.

Reviewer 3 Report
Comments and Suggestions for Authors
In my opinion the novelty of the research is not sufficient for publication in IJMS. Not only photosensitizing properties of emodin are well-known, it was applied for photodisinfection previously (10.1016/j.chemosphere.2022.135401). This effect was demonstrated for S.aureus, but ROS-mediated inactivation is expectedly non-selective and photogenerators of ROS are known to eradicate any microorganisms. Therefore, ability of emodin to inactivate Aeromonas hydrophila is quite predictable on the basis of previous findings.
Author Response
Dear editor,
Thank you for considering our manuscript entitled “Natural sunlight-mediated emodin photoinactivation of Aeromonas hydrophila” for the publication in International Journal of Molecular Sciences. We greatly appreciate the comments and suggestions from the referees and the editors, which inspired us to further explore our investigation. As to some questions raised by the reviewers, we have the following replies and the replies were highlighted in blue. We also checked the grammar and typos carefully and revised our manuscript according to the reviewers’ opinions. The revised manuscript with changes using red highlights for easy identification was also prepared.
To Reviewer 3:
Q: In my opinion the novelty of the research is not sufficient for publication in IJMS. Not only photosensitizing properties of emodin are well-known, it was applied for photodisinfection previously (10.1016/j.chemosphere.2022.135401). This effect was demonstrated for S.aureus, but ROS-mediated inactivation is expectedly non-selective and photogenerators of ROS are known to eradicate any microorganisms. Therefore, ability of emodin to inactivate Aeromonas hydrophila is quite predictable on the basis of previous findings
A: We would like to thank the reviewer for careful and thorough reading of this manuscript and for the thoughtful comments and constructive suggestions, which help us to improve the quality of this manuscript. Although the ability of emodin to inactivate Aeromonas hydrophila is predictable on the basis of previous findings, the photoinactivation of A. hydrophila is uncertain. Just like cercosporin (a natural product secreted by Cercospora), in our previous work, it can photoinactivate S.aureus effectively (Wu, et al. J Hazard Mater, 2021.), but the photoinactivation of E.coli has little effect. This study ingeniously utilized the photosensitivity of the active ingredient emodin in traditional Chinese herb, which can efficiently achieve the photoinactivation of A. hydrophila under visible light irradiation. This study highlights the potential use of emodin in sunlight-mediated applications for bacterial control, thereby providing new possibilities for the use of Chinese herbal medicine in aquatic diseases preventions.
